# The Use of Questionnaires in Pain Assessment during Orthodontic Treatments: A Narrative Review

**DOI:** 10.3390/medicina59091681

**Published:** 2023-09-18

**Authors:** Vitale Marina Consuelo, Falzinella Chiara, Sfondrini Maria Francesca, Defabianis Patrizia, Scribante Andrea

**Affiliations:** 1Unit of Orthodontics and Pediatric Dentistry, Section of Dentistry, Department of Clinical, Surgical, Diagnostic and Pediatric Sciences, University of Pavia, 27100 Pavia, Italy; marinaconsuelo.vitale@unipv.it (V.M.C.); Francesca.sfondrini@unipv.it (S.M.F.); andrea.scribante@unipv.it (S.A.); 2Department of Surgical Sciences, C.I.R. Dental School, Section of Pediatric Dentistry, University of Turin, 10124 Turin, Italy; patrizia.defabianis@unito.it; 3Unit of Dental Hygiene, Section of Dentistry, Department of Clinical, Surgical, Diagnostic and Pediatric Sciencies, University of Pavia, 27100 Pavia, Italy

**Keywords:** pain perception, discomfort, orthodontic appliances, visual analogue scale (VAS)

## Abstract

Pain is a complex multidimensional feeling combined with sensorial and emotional features. The majority of patients undergoing orthodontic treatment report various degrees of pain, which is perceived as widely variable between individuals, even when the stimulus is the same. Orthodontic pain is considered the main cause of poor-quality outcomes, patients’ dissatisfaction, and lack of collaboration up to the interruption of therapy. A deep understanding of pain and how it influences a patient’s daily life is fundamental to establishing proper therapeutic procedures and obtaining the correct collaboration. Because of its multifaced and subjective nature, pain is a difficult dimension to measure. The use of questionnaires and their relative rating scales is actually considered the gold standard for pain assessment. Choosing the most appropriate instrument for recording self-reported pain depends on a patient’s age and cognitive abilities. Although several such scales have been proposed, and a lot of them are applied, it remains uncertain which of these tools represents the standard and performs the most precise, universal, and predictable task. This review aims to give an overview of the aspects which describe pain, specifically the pain experienced during orthodontic treatment, the main tool to assess self-perceived pain in a better and more efficient way, the different indications for each of them, and their correlated advantages or disadvantages.

## 1. Introduction

During orthodontic treatment, the feeling of discomfort and the experience of some degree of pain are commonly reported [1]. Orthodontic patients may go through feelings of tension, pressure, and soreness of teeth regarding the perception of pain [2].

Pain is the main concern of people undergoing orthodontic treatment. It has been reported that the fear of pain is one of the primary factors that negatively influences a patient’s willingness for treatment uptake by discouraging it, together with the appearance of braces and the fear of being teased [3].

This may also affect their collaboration during all treatment times, the outcome, patient satisfaction, and even their quality of life [4]. Patients’ dissatisfaction may lead to longer treatment times and poor-quality therapy results [5]. A significant proportion of patients (8%) decide to interrupt the therapy because of the pain suffered in the initial stages of treatment [6].

For these reasons, orthodontic pain, its management, and assessment are fundamental for patients and clinicians who need to handle the situation in the best possible way [5].

Applying an assessment tool is necessary to evaluate some specific features. Every tool must have the characteristics of validity and sensitivity [7].

Validity is the ability of an assessment to measure what it was supposed to measure. In the case of a questionnaire, the validity permits the collection of high-quality data with increasing credibility. To be sure that the questionnaire is valid and provides the information for which it is developed, it must undergo a validation process [8]. A valid questionnaire must have the following aspects: (1) simplicity and viability, (2) reliability and precision, (3) be adequate for the problem it is intended to measure, (4) reflect the underlying theory or concept to be measured, (5) and be capable of measuring change [9].

Reliability indicates low sensitivity to random error. Reliability is linked to reproducibility: the test provides the same results even if repeated. The third fundamental feature of an assessment tool and questionnaire is sensitivity, especially if it is used for evaluative purposes. This could underline changes in the same patient during the therapy [7].

## 2. Materials and Methods

### 2.1. Focused Questions

What are the most relevant aspects when assessing pain, especially chronic pain? What are the main tools used to evaluate it?

### 2.2. Eligibility Criteria

The following inclusion criteria guided the analysis of the studies: (I) absence of Ethics Committee approval, (I) study design—clinical trials, case–control studies, case reports, (II) participants—patients with painful conditions and/or psychological implications, (III) interventions—pain assessment (IV), outcome—pain intensity, pain assessment tools, and psychological assessment tools. We only included studies with full text. Only studies that met all the inclusion criteria were included. We also excluded: (I) abstracts of articles published in non-English languages, (II) duplicate studies, (III) in vitro or animal clinical studies, (IV), non-pertinent studies, and (V) study design—narrative reviews, systematic reviews.

### 2.3. Search Strategy

The PICO model (Population, Intervention, Comparison, Outcome) was used to perform this review through a literature search of the PubMed (MEDLINE) and Scopus electronic databases. Abstracts of studies that evaluated the assessment of pain and/or psychological implications were reviewed.

### 2.4. Research

The medical subject heading (MeSH) terms are pain, pain assessment, questionnaire, visual analogue scale, and orthodontic. An electronic search was carried out using the PubMed (MEDLINE) and Scopus databases. Articles published in the years 1962–2022 were targeted. The duration of data extraction was about 8 weeks. The last search was performed on 29 July 2023. All the titles and abstracts were read thoroughly from the articles searched primarily, and nonrelevant studies were excluded. The relevant articles were enlisted and scrutinized for any similar studies that matched our inclusion criteria. For the extraction of pertinent results, we read the full texts of the included studies, and the findings were recorded.

## 3. Results

The primary search identified 883 articles based on the MeSH terms, published from 1983 to 2021. Following these, the research was restricted to narrative reviews, systematic reviews, case reports, and meta-analyses concerning human studies and the English language, and, in total, 96 results were found. As a result, it was decided that only articles with full text should be used; therefore 38 articles were screened and evaluated for eligibility (Figure 1).

Specifically, the selected articles employed questionnaires to collect data concerning both the quality and quantity of pain during different orthodontic treatments. These were related to fixed appliances (self-legating, different arch wires, mini-implants, rapid maxillary expanders) as well as Invisalign ones.

## 4. Discussion

### 4.1. Features of Pain

In 2020, the IASP (International Association for the Study of Pain) updated the last definition of pain: “An unpleasant sensory and emotional experience associated with or resembling that associated with actual or potential tissue damage”. The association adds six features that better characterize pain: (1) pain is a personal experience, influenced by biological, psychological, and social factors, (2) pain and nociception are different concepts, with the first being much more complex than the second, (3) individuals learn what pain means during their life, (4) when a patient says they are feeling pain, their statement must always be respected, (5) despite its protective and adaptive functions, pain can have adverse effects on psychological and social functioning, and (6) verbal description is only one of the methods for depicting pain [10].

Based on this definition, pain is a multidimensional feeling that can be described as a union of two components: sensorial and emotional. The sensorial one is derived from anatomical features of the nervous system: stimuli are received and transported to the central nervous system (CNS) through a tri-neuronal network that originates from the periphery and reaches the cerebral cortex. The emotional component is related to the limbic system, a group of structures located in the deep telencephalon, which is liable for the subjective part of the painful stimulus [11].

### 4.2. Features of Orthodontic Pain

Tooth movement is only possible through a process of inflammation as an outcome of the forces applied during the course of orthodontic therapy [12].

The feeling of dental pain is a consequence of stretch and compression forces applied to the periodontal ligament and the alveolar bone. This causes a change in the blood flow of the periodontal apparatus to induce the release of the tissue biochemical moderators and determine the inflammatory reaction, which leads to the sensation of pain [8].

Orthodontic patients experience pain during treatment in a range of 70–95%, and they report the highest intensity of pain during the first 24 h [13,14,15].

According to the literature, 39–49% experience pain during the whole course of treatment or after the removal of the appliance [16,17].

The intensity of pain depends on several factors, both subjective and objective: gender, age, sex, the magnitude of force applied, state of emotion and stress, personal pain threshold, individual culture, and previous pain experience [1,14,18,19].

Furthermore, the type and intensity of suffering can change depending on the appliances used and the type of force applied [20].

Although the quantity and quality of pain may depend on the typology of appliances, it has been stated that 70% of orthodontic patients experienced some type of suffering or pain, regardless of the type of appliance used [21].

In the case of fixed-appliance-based treatment, within two hours from the insertion of the first arch wire, patients reported a feeling of pain that increased progressively. After 24 h, almost the totality of patients (95%) reported pain that reached its peak and had started to decrease [16,22].

Burstone classified pain into three degrees: (1) pain is produced by heavy pressure on the tooth with an instrument in the same direction as the force of the fixed appliance, (2) pain evocates due to clenching or heavy biting, and (3) the patient suffers spontaneous pain or is unable to masticate food [12].

Bergus et al. described two different types of pain experienced during the active orthodontic phase of treatment: constant pain and chewing-related pain. In total, 82% of orthodontic patients report pain during chewing/biting, while 24% declare constant pain on the first day after installation. The suffering decreases gradually from the second day to the seventh when the spontaneous pain reported is null. However, 30% still reported pain during chewing/biting [1].

A large number of unidimensional and multidimensional assessment tools were suggested to measure patient pain. One-dimensional tools focus on and consider only one characteristic aspect of pain (such as pain intensity), simplifying the pain experience. These instruments are rapid to administer and require less intellectual ability in comparison to multidimensional ones [23].

### 4.3. Pain Intensity Assessment

The European Palliative Care Research Collaborative agrees that the most clinically relevant dimension to pain assessment, regardless of disease or condition, is pain intensity (PI) [24].

The extent of pain intensity is conventionally assessed subjectively using different types of pain scales and unidimensional tools [25].

Over the years, several subjective methods, which include pain scales, like NRS, VAS, and VRS, have been proposed to assess the experience of pain more accurately [26]. Currently, these scales are the best instruments for measuring individual pain and pain assessment tools based on self-reported perception using questionnaires, and their relative scales are the gold standard [24].

A preponderance of evidence demonstrates that the 100 mm Visual Analog Scale (VAS) is by far the most frequently used assessment pain tool [24].

This scale is composed of a line of a fixed length with defined endpoints. One extremity of the line is labeled “no pain” and the other end is “worst pain imaginable” (Figure 2). Patients are required to indicate the place on the line corresponding to their self-perceived state [27].

Patients mark, on the 10 cm line, the point that represents their self-perception to quantify and record the pain intensity in a range from 0 to 100 mm. The obtained value provides a quantitative variable of pain [25].

The VAS is a popular tool in the clinical setting because of its rapidity of administration and simplicity. It can be used by anyone cognitively capable to respond to clinician instructions [28]; however, some patients can find the tool too difficult or too abstract to understand, especially patients with a physical disability or cognitive dysfunction [23]. One of the main limitations of the VAS is that it must be completed on paper or electronically; it cannot be completed orally [29].

The VAS has shown good reliability and sensitivity, being able to detect PI variation over time [30,31].

The VAS scale is frequently used to assess pain intensity during treatment in the orthodontic field.

A clinical trial performed on 64 patients was aimed at comparing the pain intensity between subjects treated with passive self-ligating fixed appliances and those treated with Invisalign aligners. The subjective perception of pain was recorded using a questionnaire based on the VAS [32].

The same results were obtained in a prospective longitudinal study, which was performed to make a comparison between Invisalign and fixed appliances during the first week of therapy. It was shown that patients treated with fixed appliances reported a higher level of pain intensity, according to VAS, compared to those treated with Invisalign therapy [33].

Another study was carried out to make a comparison between patients treated using three different orthodontic appliances in terms of pain level: clear aligners, self-ligating, and conventional. During the first week of therapy, the VAS scale demonstrated that the lowest level of pain was reported by the clear aligner group. Furthermore, the self-ligating group reported a significantly lower level of pain compared with the conventional one [34].

For an intensity pain assessment during adulthood or childhood, the Numerical Rating Scale (NRS) can be used. The NRS is a segmented numerical version of the VAS, which consists of a numerical scale ranging from 0 to 100, where the extreme 0 corresponds to “no pain” and the other extreme represents “pain as bad as it could be” (Figure 3). Patients are required to mark the number corresponding to their perceived level of pain intensity [35].

The intensity of pain using the NRS is assessed as follows: no pain = 0, mild pain = 1–3, moderate pain = 4–6, and severe pain = ≥ 7 [36].

Using the NRS, a study performed on 165 patients who agreed to have a miniscrew insertion during their orthodontic treatment to facilitate tooth movement was aimed at evaluating the experience of pain during the post-operatory time [37].

Although the NRS and the VAS are the most widely used self-reported pain scales, they have a disadvantage correlated to the request of a notable level of abstract thinking to correlate pain experience with a number or a point on the line [38].

The VRS is more appropriate for patients with cognitive impairment thanks to its simplicity. It was shown that the VRS provides a higher response compared with the NRS and VAS [39].

The VRS is made up of a list of adjectives corresponding to different levels of pain. The first adjective is related to lower pain intensity, while the others correspond to a gradually higher pain level. Examples of descriptors that can be used are “none”, “mild”, “moderate”, and “severe” [40].

Among the various advantages of the VRS, its rapidity and simplicity of use can be underlined. The VRS is easy to teach to patients with regard to its correct compilation, and it is easy to score and document the results. Furthermore, the tool is well validated and can be used to detect treatment effects. The verbal rating scale overcomes some limitations of the VAS as it can be administered verbally, thereby permitting its use by those who have physical or visual difficulty [23].

In the literature, among the orthodontic studies which take advantage of VRS, a prospective randomized trial was performed to compare pain experience during orthodontic treatment with self-ligating brackets and conventional ones [41].

### 4.4. Pain Assessment in Children

Pain assessment in children is a challenge linked to the complex nature of pain, language limitation and difficulty in comprehension and self-reporting, and the social feature of pain (different pain experiences depending on age, sex, and race) [42].

The level of cognitive development, language comprehension, and vocabulary may influence a child’s capacity to provide a self-report of pain intensity and their ability to use a pain scale accurately [43,44].

Children understand and express pain differently depending on their developmental stage. Thus, the pain assessment must be age dependent [45].

Pawar and Garten described the difference in the expression of pain in various pediatric age groups:

Preschoolers may verbalize the intensity of pain, see pain as punishment, exhibit thrashing of arms and legs, attempt to push a stimulus away before it is applied, be uncooperative, need physical restraint, request emotional support, understand that there can be secondary gains associated with pain, or be unable to sleep.

School-aged children may verbalize pain, use an objective measurement of pain, be influenced by cultural beliefs, experience nightmares related to pain, exhibit stalling behaviors, have muscular rigidity, e.g., clenched fists, white knuckles, gritted teeth, and contracted limbs, exhibit body stiffness, closed eyes, wrinkled forehead, engage in the same behaviors listed for preschoolers/young children, or be unable to sleep.

Adolescents may localize and verbalize pain, deny pain in the presence of peers, have changes in sleep patterns or appetite, be influenced by cultural beliefs, exhibit muscle tension and body control, display regressive behavior in the presence of family, or be unable to sleep [23].

Nevertheless, children aged ≤7 are able to report pain severity, and from age 3 to 7 they progressively increase their capacity to describe, analyze, and localize it [36].

In the pediatric field, facial scales are a widespread tool for pain severity assessment. They consist of a sequence of facial expressions that correspond to a spectrum of increasing pain intensity. Although various face scales have been developed, the Wong–Baker Faces Pain Rating Scale (WBS) has been demonstrated as appropriate in various pediatric settings and is the most widely applied [36,46].

The Wong–Baker Faces Pain Scale is composed of six faces, where the first one is a happy smiling face and the last one is sad and tearful. The faces in the middle represent varying and increasing degrees of pain. The patient is asked to choose the face most like his/hers during the pain [47].

This tool is simple and quick to administer, is easy to score, requires no reading or verbal skills, is unaffected by issues of gender or ethnicity, and provides three scales in one (i.e., facial expressions, numbers, and words). For these reasons, the most widely used scales for children’s pain intensity records are the Wong–Baker Faces Pain Rating Scale, various visual analog scales, and verbal numeric scales [43].

The leading disadvantage of this tool is the possibility of measuring mood instead of pain, and the interpretation of sad or crying faces is dependent on different cultures [23].

Nowadays, one of the most widely used instruments in school-age patients and adolescents to obtain self-reports of pain intensity is the Verbal Numerical Rating Scale (NRS or VNRS). This scale is made up of a list of descriptors that correspond to an increase in pain intensity. The patient has to choose one of six descriptors (“No pain”, “Mild pain”, “Moderate pain”, “Severe pain”, “Very severe pain”, and “Worst pain possible”) that best represents the level of self-perceived pain intensity (Figure 4). The numbers may be used to ease the recording of the results. The VNRS has a marked validity and reliability in young patients aged ≥8 years [48,49].

Because of lower developmental skills, children aged ≤7 years lack the developmental skills necessary for the accurate use of this pain tool [50,51].

In a recent prospective study performed in 2020, a questionnaire composed of the NRS and the Wong–Baker Faces Pain Scale was used to assess and compare the intensity of pain following the maxillary expansion with two different types of expanders (Hyrax and Haas type). The conclusions suggested that the pain was reported in both types of expansion, and the highest pain was shown on the first day of activation in the Hyrax group [52].

Another recent study based on VAS and the Wong–Baker Faces Pain Rating Scale was carried out (Figure 5). The study recorded the self-pain perception experienced by patients after a mini-implant placement, concluding that the maximum pain was recorded in the case of a palatal miniscrew [53].

Since pain is defined as a multidimensional emotion, a multidimensional tool permits a much more comprehensive and exhaustive pain assessment. In this way, not only PI but all the pain features, including psychological, emotional, and behavioral ones, are examined [54].

The main disadvantage of the traditional methods of pain measurement (visual analogue scale, numerical rating scale, verbal rating scale) is that they treat pain as if it was a one-dimensional experience that can be described using only the dimension of the intensity measured with a single unique scale. Although intensity is one of the main features of pain, the word “pain” refers to several qualities categorized under a single linguistic term and not to a specific, single feeling that varies only in intensity [55].

The McGill Pain Questionnaire (MPQ) is a multidimensional tool developed by Melzack and Torgerson for a self-reported measure of pain assessment, evaluating both its quality and quantity aspects. This questionnaire permits the estimation of the quality dimension by considering pain intensity, localization, temporality, and affective aspects of pain. To fulfill these tasks, the original MPQ comprised 78 pain descriptors categorized into four major classes. The classes included terms that describe the sensory, affective, evaluative, and miscellaneous quality of the pain experience [56].

The MPQ is also employed to assess pain intensity employing the Pain Rating Index (PRI) and the Present Pain Intensity (PPI). The PRI is based on a 1–5 intensity scale, and it is built thanks to a numerical grading of words describing the sensory, affective, evaluative, and miscellaneous features of pain. Patients are instructed to select only those descriptors that accurately depict their sensations at that moment. Within this scoring system, the word within each subclass that signifies the least pain is attributed a value of 1, followed by a value of 2 for the subsequent word and so on. The rank values of the word chosen by the patient are summed to obtain a separate score for the four major classes and an overall score (Figure 6) [57]. In addition to the PRI, the PPI score, which ranges from 1 (mild) to 5 (excruciating), is considered. It can be obtained by scoring a response to the question “Which word describes your pain right now?” [58].

Over the years, the MPQ has become one of the most commonly widespread clinical and research tools. This instrument is not perfect, and numerous variants have been proposed [59,60].

The MPQ takes 5–10 min to administer, which is too long for research that requires a more rapid collection of data. With this in mind, an alternative and shortened form of the standard MPQ has been provided. The short-form MPQ (SF-MPQ), which permits obtaining the needed data in a limited time, was proposed in 1987 by Melzack and Ronald. The SF-MPQ is based on the selected and most representative words from the three categories of the original questionnaire. The magnitude of the pain intensity is assessed using the Present Pain Intensity (PPI) and the Visual Analog Scale [61].

A study, carried out on a sample of 189 orthodontic patients over a range of 12–30 years, took advantage of the submission of the modified McGill Pain Questionnaire with VAS to assess the quality and intensity of pain after the initial insertion of six different NiTi orthodontic wires as a part of fixed appliances. Each patient received one of the six different superelastic or heat-activated NiTi arch wires to solve the crowding and level of the teeth in the first phase of the therapy. The research concluded that no correlation was shown between the patient’s pain perception and the different types of NiTi arch or the gravity of the crowding, even though most of the patients reported the pain as starting 12 h after the application and lasting for the first 3–4 days [62].

A short version of the MPQ was applied to study the effect of orthodontic pain on the Quality of Life (QoL) in patients undergoing orthodontic therapy. In total, 200 patients aged from 13 to 18 were enrolled and underwent the McGill short-form questionnaire with VAS and PPI to evaluate the intensity and severity of pain and if it influenced their quality of life during the first month after the insertion of the appliance. The results demonstrated that orthodontic fixed appliances have a considerable impact on patients’ daily life, especially during the first few days, when 85% of them think that their QoL is reduced as a consequence of the beginning of the therapy [63].

This report exhibits some limitations. The search procedure could have been too specific for a scoping question. Moreover, the definition of the optimal pain assessment scale is complex due to the wide array of available tools that have undergone testing.

Another limiting factor consists of the choice of evaluating articles solely in the English language as an inclusion criterion.

## 5. Conclusions

The multidimensional subjective nature of pain makes its objective measurement difficult. Pain assessment is a complex process that aims to describe a multifaced phenomenon consisting of different aspects, sensorial, psychological, and experiential, in an objective way. Various assessment tools have been developed and used in the orthodontic field for sensorial aspect assessment based on rating scales and questionnaires: VAS, NRS, VRS, and MPQ.

The use of these instruments makes comparisons between the pain experienced during different orthodontic therapies or using different appliances, the assessment of the quality and quantity of pain, and the monitoring of pain possible.

## Figures and Tables

**Figure 1 medicina-59-01681-f001:**
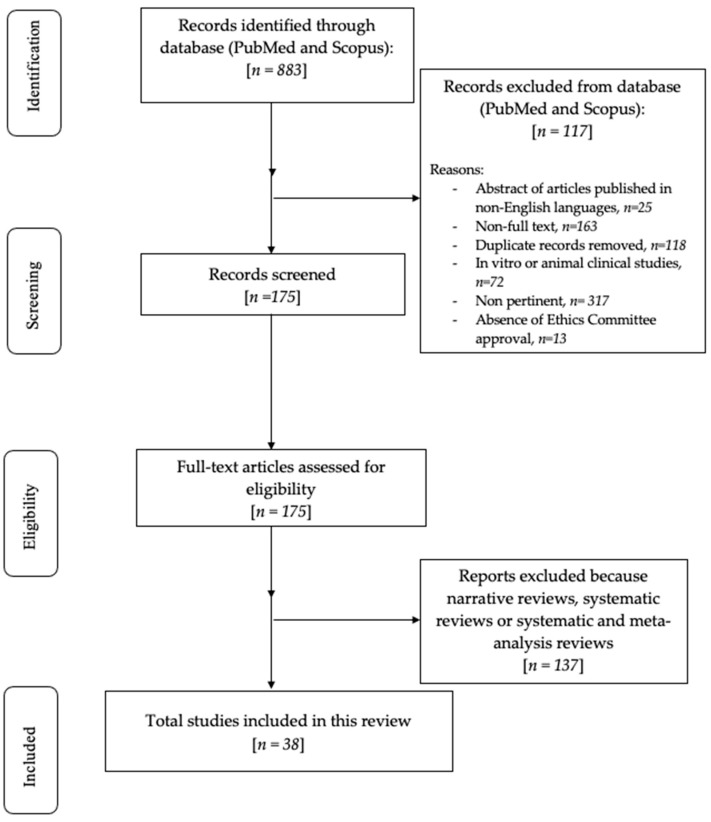
Flowchart of the review process.

**Figure 2 medicina-59-01681-f002:**
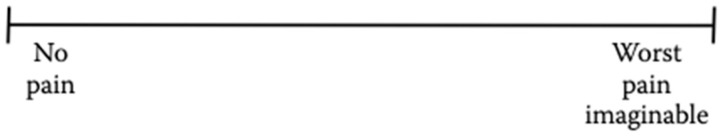
The Visual Analog Scale.

**Figure 3 medicina-59-01681-f003:**
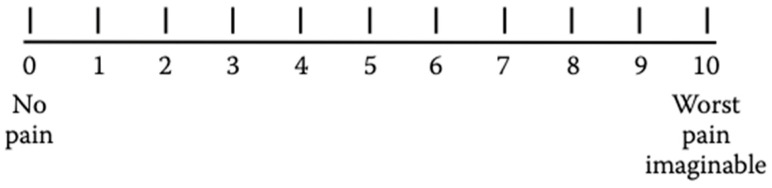
Numerical Rating Scale (NRS).

**Figure 4 medicina-59-01681-f004:**
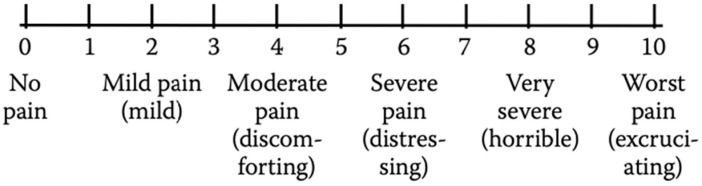
Verbal Numerical Rating Scale (VNRS).

**Figure 5 medicina-59-01681-f005:**
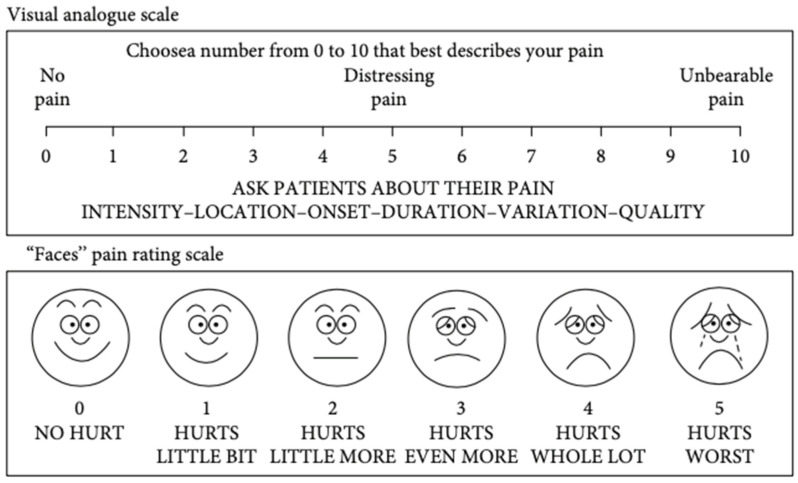
Visual Analogue Scale and Faces Pain Rating Scale [53].

**Figure 6 medicina-59-01681-f006:**
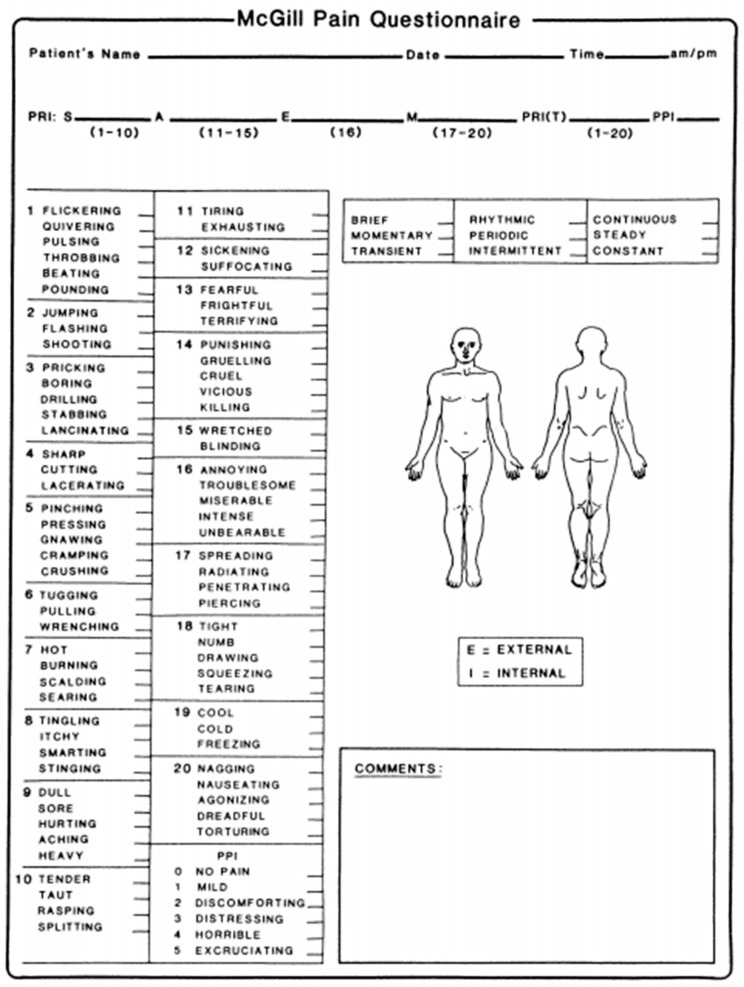
The McGill Pain Questionnaire: the sensitive (1–10), affective (11–15), evaluative (16), and miscellaneous (17–20) groups and their descriptors. The Pain Rating Index and the Present Pain Intensity [57].

## Data Availability

Data supporting the reported results are available on request from the corresponding authors.

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
