# Peer review of "The Use of Questionnaires in Pain Assessment during Orthodontic Treatments: A Narrative Review"

_medicina, 2023, doi:10.3390/medicina59091681_

Round 1
Reviewer 1 Report
RESULTS:
“Only articles with full text or free full text should be used”: please explain if this sentence means that only open access articles were included in this review.
“absence of Ethics Committee approval” is reported in Figure 1 as a reason to exclude 13 manuscripts from records identified: this exclusion criteria should be declared also in the Materials & Methods section. In the same figure, is reported “reports excluded because narrative reviews, systematic reviews or systematic and meta-analysis review”, but in the Materials and Methods section narrative and systematic reviews are mentioned as inclusion criteria: please check this and modify the text or the figure accordingly.
DISCUSSION:
The visual analog scale should be Figure 2, because flowchart of the review process was Figure 1.
“Figura 2” should be “Figure 3”.
The meaning of the abbreviation VRS should be added in the manuscript the first time it is used.
“Figura 3” should be “Figure 4”.
“and the Wong-Baker Faces Pain Rating Scale was carried out (Fig. 1).” Should be “and the Wong-Baker Faces Pain Rating Scale was carried out (Figure 5).”
“Figure 4 - Visual Analogue Scale and Faces Pain Rating Scale.” should be “Figure 5 - Visual Analogue Scale and Faces Pain Rating Scale”.
“treat pain as if it were a single” should be “treat pain as if it was a single”.
“aged from 341 13 to 18 are enrolled and” should be “aged from 341 13 to 18 were enrolled and”
The decision to include only studies with free full text available and to exclude abstracts of articles published in non-English languages is a limitation of the study that should be discussed in this section.
CONCLUSIONS:
The meaning of the abbreviation FPS should be added in the manuscript the first time it is used.
Few typo errors
Author Response
I have implemented the suggested corrections and added the required sections.

Reviewer 2 Report
This manuscript discusses the assessment of pain in the context of orthodontic treatment. Here's a critical appraisal of the various sections of the manuscript:
1. Introduction:
- The introduction sets the stage by highlighting the importance of pain assessment in orthodontic treatment.
- It effectively presents the problem, the significance of the issue, and its impact on treatment outcomes.
- The references cited in this section provide relevant background information to support the statements made.
2. Materials and Methods:
- The focused questions and eligibility criteria are clearly defined, helping readers understand the scope of the study.
- Using the PICO model for search strategy is appropriate for structuring the literature search.
- Inclusion and exclusion criteria are well-defined, providing transparency about the selection process.
- A flowchart depicting the review process is a good visual aid for understanding the study selection.
3. Results:
- The section presents a concise summary of the search and screening process.
- Using a figure (Flowchart) is beneficial in clearly illustrating the study selection process.
- However, the results section is relatively brief and could provide more information on the characteristics of the included studies.
4. Discussion:
- This section provides a comprehensive discussion of various aspects of pain assessment, including pain features, orthodontic pain, pain intensity assessment, and pain assessment in children.
- The discussion of the features of pain, orthodontic pain, and pain intensity assessment is detailed and supported by relevant citations.
- Using figures to illustrate pain assessment tools (Visual Analog Scale, Faces Pain Rating Scale, etc.) is helpful.
- The discussion on pain assessment in children is particularly informative, addressing the challenges and appropriate tools for this population.
- The section effectively compares pain assessment tools and provides examples of studies using these tools.
- However, the section on McGill Pain Questionnaire (MPQ) could benefit from clearer organization and a more focused discussion.
5. Conclusions:
- The conclusions summarize the key points discussed in the manuscript.
- It underscores the importance of pain assessment tools in orthodontic treatment.
- The discussion throughout the manuscript supports the conclusions made in this section.
Overall Evaluation:
- The manuscript provides a comprehensive overview of pain assessment in the context of orthodontic treatment.
- It effectively outlines various pain assessment tools, their advantages, and limitations.
- The manuscript is well-structured, with each section logically flowing into the next.
- Using figures and references adds depth and credibility to the discussion.
- However, some sections, such as the McGill Pain Questionnaire discussion, could benefit from clearer organization and more focused content.
Suggestions:
- Provide more detailed information about the characteristics of the included studies in the Results section.
- Consider reorganizing and refining the discussion of the McGill Pain Questionnaire (MPQ) for better clarity.
- If possible, provide more recent references to strengthen the currency of the information.
Overall, the manuscript appears well-researched and informative, covering a wide range of relevant topics related to pain assessment in orthodontic treatment.
The English language in the manuscript is generally clear and understandable. The manuscript appropriately uses technical terms related to orthodontics and pain assessment, suggesting a good level of domain-specific language usage. However, there are some areas where the language could be further refined for clarity and readability:
1. Sentence Structure and Length: While the manuscript uses complex sentence structures, some sentences could be broken down into smaller, more concise sentences to enhance readability.
2. Word Choice and Flow: There are instances where the choice of words or sentence flow could be improved to enhance coherence and logical progression of ideas.
3. Grammar and Punctuation: grammar and punctuation are acceptable, but careful proofreading could help eliminate any minor errors.
4. Clarity in Explanation: Some concepts, especially in the "Features of orthodontic pain" section, could benefit from clearer explanations for readers who might not be familiar with orthodontic terminology.
5. Organization of Content: The McGill Pain Questionnaire (MPQ) discussion could be organized more effectively to present information in a clearer, more structured manner.
In any case, careful proofreading or review by a language editor could help polish the language and enhance the overall readability and impact of the manuscript.
Author Response
|
Thank you very much for taking the time to review this manuscript. Please find the detailed responses below and the corresponding revisions/corrections highlighted/in track changes in the re-submitted files. |
